# Differences in COVID-19 Vaccine Acceptance, Hesitancy, and Confidence between Healthcare Workers and the General Population in Japan

**DOI:** 10.3390/vaccines9121389

**Published:** 2021-11-24

**Authors:** Megumi Hara, Motoki Ishibashi, Atsushi Nakane, Takashi Nakano, Yoshio Hirota

**Affiliations:** 1Department of Preventive Medicine, Faculty of Medicine, Saga University, Saga 849-8501, Japan; 2Clinical Epidemiology Research Center, Medical Co. LTA, Fukuoka 813-0017, Japan; motoki-ishibashi@lta-med.com (M.I.); atsushi-nakane@lta-med.com (A.N.); hiro8yoshi@lta-med.com (Y.H.); 3Department of Pediatrics, Kawasaki Medical School, Okayama 701-0192, Japan; ndhkk029@ybb.ne.jp

**Keywords:** COVID-19 vaccine, vaccine hesitancy, vaccine acceptance, health literacy, immunization

## Abstract

Little is known about the differences in coronavirus disease (COVID-19) vaccine acceptance and hesitancy between the general population and healthcare workers in Japan. To compare these differences, a nationwide web-based cross-sectional survey was conducted on 19 January 2021, shortly before the initiation of COVID-19 vaccinations in Japan. A total of 6180 men and women aged 20–69 years and 1030 healthcare workers aged 20–69 years were enrolled. Data on COVID-19 vaccine acceptance, basic characteristics, including socioeconomic factors, and confidence in immunization in general were collected. COVID-19 vaccine acceptance was also evaluated under hypothetical vaccine effectiveness and adverse event frequencies. Factors associated with vaccine hesitancy were examined using multinomial logistic regression analysis. The COVID-19 vaccine acceptance rate was 48.6% among the general population and was lower among nurses (45.5%) and medical clerks (40.7%). Women and young adults had significantly higher COVID-19 vaccine hesitancy odds ratios, and current smokers had significantly lower odds ratios. The frequency of adverse events was a COVID-19 vaccine hesitancy factor. Even if these factors were adjusted, COVID-19 vaccine hesitancy among nurses was 1.4 times higher than that among the general population. Thus, interventions to improve health literacy and vaccine hesitancy among the general population and healthcare workers, especially nurses, are needed.

## 1. Introduction

Vaccine hesitancy is a challenge in promoting vaccination against coronavirus disease (COVID-19). In 2015, the World Health Organization (WHO) Strategic Advisory Group of Experts on Immunization defined vaccine hesitancy as a delay in acceptance or refusal of vaccination despite the availability of vaccination services [1]. Vaccine hesitancy is complex in the context in which it occurs and varies by region, era, and vaccine type. Hesitancy factors include individual knowledge and beliefs, confidence in vaccines and public health, and the convenience of vaccination [2]. A COVID-19 vaccine acceptance (willingness to be vaccinated) survey conducted in Europe in April 2020 showed approximately 70% acceptance of the vaccine among respondents. Furthermore, among men, acceptance increased with age. Adverse events and concerns about safety were cited as hesitancy factors [3]. A survey conducted in the United States in May of the same year also indicated approximately 70% acceptance of the COVID-19 vaccine [4]. In a survey of 13,426 people in 19 countries in June of the same year, approximately 70% of the respondents were willing to receive the COVID-19 vaccine. Furthermore, the intention to receive the vaccine increased with confidence in information from governments [5]. Since then, a number of studies have been conducted on vaccine acceptance and hesitancy [6]. The duration since the start of the pandemic, sex, age, educational and socioeconomic factors, region, religion, trust in the government, characteristics such as confidence in the vaccine, and recommendation of vaccination from healthcare workers have been reported to be impactful factors related to the intention to receive COVID-19 vaccines [6]. In addition, several studies have evaluated vaccination intention and hesitancy among healthcare workers [7,8,9]. Vaccine acceptance was higher among medical personnel attending patients with COVID-19 [7]. Furthermore, vaccine acceptance was higher among male healthcare workers, older healthcare workers, and doctors [8]. A recent meta-analysis reported a lower (51%) COVID-19 vaccine acceptance rate among healthcare workers, especially among female healthcare workers [9].

A pre-COVID-19 survey of more than 280,000 people in 149 countries and regions found Japan to have one of the lowest rates of vaccine confidence in the world [10]. Thus, COVID-19 vaccine hesitancy in Japan may be higher than that in other countries. Therefore, it is important to investigate vaccination acceptance and hesitancy and to clarify the related factors in Japan as the country promotes vaccination. To date, several surveys have been conducted on COVID-19 vaccine acceptance and hesitancy in Japan [11,12,13,14]. These surveys show that COVID-19 vaccine acceptance among the Japanese general population is approximately 50–70%. Similar to Western countries, vaccination acceptance is associated with male sex, old age, underlying diseases, high socioeconomic status, and recommendations from healthcare workers. However, no studies have compared vaccine acceptance and hesitancy between healthcare workers and the general population in Japan. Investigating vaccine acceptance and hesitancy between healthcare workers and the general population in Japan might have lots of relevant lessons for developed and developing countries in terms of how to monitor or programmatically approach vaccine hesitancy.

This study aimed to compare COVID-19 vaccine acceptance and hesitancy between the general population and healthcare workers in Japan.

## 2. Materials and Methods

### 2.1. Study Sample and Data Collection

This was an Internet-based cross-sectional survey study conducted in Japan. The survey was conducted on 19 January 2021, at a time when the number of COVID-19 cases had significantly increased, and a state of emergency was declared for Tokyo, Chiba, Saitama, Kanagawa, Tochigi, Gifu, Aichi, Kyoto, Osaka, Hyogo, and Fukuoka.

Study participants were registered on a panel of a web survey company (Macromill Inc., Tokyo, Japan). Panel membership was provided on a voluntary basis, and incentives were provided as points when the participants joined the survey. These points could be used to purchase products and services from partner companies. Approximately 1.2 million people were registered on this research panel.

We calculated the sample size needed to investigate factors associated with vaccine hesitancy as follows: assuming α = 0.05/number of variables (50 items) = 0.001, β = 0.20, odds ratio = 1.5, vaccine hesitancy = 30–50%, and the proportion of associated factors was 10–20%, the required sample size was 2000–5000. We set 6000 men and women aged 20–79 years from the general population and 1000 healthcare workers aged 20–69 years as the target study population who were extracted from the membership panel. The survey was explained by e-mail, and study participants were enrolled using e-mails and apps until the target number was reached. The questionnaires were placed in a secure section of the website, and responders were required to answer each question so that there were no missing variables. Participants were considered to have consented by answering the survey questions. This study was conducted in accordance with the guidelines of the Declaration of Helsinki and approved by the Ethics Committee of Saga University, Saga, Japan (No: R2–24).

### 2.2. Measures

#### 2.2.1. Assessment of COVID-19 Vaccine Acceptance and Hesitancy

COVID-19 vaccine acceptance and hesitancy were measured by how many respondents agreed with the following statement: “When a vaccine for COVID-19 becomes available, I will get vaccinated.” A five-point Likert-type rating scale was used with the following range of choices: “strongly disagree, disagree, neither agree nor disagree, agree, and strongly agree.” Those who chose “strongly disagree” and “disagree” were defined as hesitant. Those who chose “agree” and “strongly agree” were defined as accepting. Those who chose “neither agree nor disagree” were defined as “not sure.” The COVID-19 vaccine acceptance levels were also measured under the following hypothetical effectiveness and adverse event frequencies: effectiveness of protection against severe symptoms, 50%, 70%, and 90%; duration of vaccine effectiveness, 1 year, 3 years, and 5 years; serious adverse events, such as death or hospitalization, occurrence in one in 10,000 doses, one in 100,000 doses, and one in 1,000,000 doses; and mild adverse events, such as flu-like symptoms or high fever, occurrence in one in two doses, one in five doses, and one in ten doses.

#### 2.2.2. Assessment of General Vaccine Confidence and Literacy

Vaccine confidence is defined as trust in the effectiveness and safety of vaccines and the healthcare system that delivers them [15]. In this study, vaccine confidence and literacy were measured using 14 questions based on a validated scale or index [16,17,18,19]. We selected six of the ten items of the Vaccine Hesitancy Scale, which was developed by the WHO Strategic Advisory Group of Experts [1] and validated by Larson et al. [20]. The items included importance, effectiveness, herd immunity, trust, and safety as represented by the following statements: vaccines are important for my health, vaccines are effective, my vaccination is important for the health of others in my community, new vaccines carry more risks than older vaccines, I am concerned about serious adverse effects of vaccines, and I do not need vaccines for diseases that are no longer common. The remaining eight items related to vaccine confidence and literacy, such as complacency (not perceiving disease as high risk); constraints (structural and psychological barriers); and calculation (engagement in extensive information searching) [16,17,21] as represented by the following statements: vaccines are safe; serious adverse events may occur due to the vaccination; I have difficulty getting immunized (no time, inconvenient location of medical institutions, and other related factors); we do not need to take voluntary vaccination; I do not take the vaccine if everyone around me is immunized; it is easy to obtain correct information on immunization; it is easy to understand why immunization is needed; and I have been able to accurately understand the vaccinations I have received. Responses were on a five-point Likert-type rating scale ranging from “strongly disagree” to “strongly agree,” as mentioned earlier. To measure vaccine confidence and literacy, we built a total score by adding the points allocated to the answers as follows: strongly disagree = 1, disagree = 2, neither agree nor disagree = 3, agree = 4, strongly agree = 5. Some items were allocated scores in reverse order.

#### 2.2.3. Factors Associated with Vaccine Acceptance or Hesitancy

Participants chose all of the factors that applied to them as important factors in deciding whether or not to vaccinate in listed factors in follows: age, comorbidity, occupation vaccination fee, doctors’ recommendation, vaccine effectiveness, duration of effectiveness, frequency of adverse events, family recommendation, and the COVID-19 epidemic situation.

#### 2.2.4. Assessment of Sociodemographic Factors

Data on the following sociodemographic factors were collected: sex, age group (20–29, 30–39, 40–49, 50–59, or 60–69 years), occupation, state of emergency in their residential area, marital status (married or not), have children or not, household annual income category (<2 million yen, 2 to <4 million yen, or ≥4 million yen), highest level of education, current smoking status, height, weight, comorbidities (any of diabetes mellitus, hypertension, pulmonary disease, heart disease, hyperlipidemia, chronic kidney disease, liver disease, immune disease, allergy, the other), and influenza vaccination status in the previous (2019/20 season) and present season (2020/21 season).

### 2.3. Statistical Analysis

A χ^2^ test was performed to test the differences between the general population and healthcare workers. A multinomial logistic regression analysis was performed to test the factors associated with COVID-19 vaccine hesitancy, with reference to the acceptance group. Among the significant factors identified by univariate analysis, those factors that were strongly correlated with one another, and were more compatible with the model, were used as explanatory variables in the multivariate analysis. Age- and sex-adjusted scores for vaccine confidence and literacy were analyzed using multiple regression analysis. Two-tailed tests were performed at a significance level of <5%. SAS version 9.4 (SAS Institute Inc., Cary, NC, USA) was used for statistical analysis.

## 3. Results

A total of 6180 men and women aged between 20 and 69 years (515 men and women per 10-year age group) and 1030 healthcare professionals participated in this study. A comparison of the characteristics of the general population and healthcare workers is presented in Table 1. The healthcare workers group had significantly higher proportions of women, younger age groups, higher annual household incomes, higher educational attainment, those who were vaccinated against influenza in the current and last seasons, and those who understood the implications of vaccine effectiveness. On the other hand, there were significantly lower proportions of people living in the declared emergency areas, obesity, and current smoking among the healthcare workers compared with the general population. Vaccination cost and recommendation from the doctor were the factors affecting vaccination among the general population. Comorbidity, occupation, vaccine effectiveness, duration of vaccine effectiveness, and frequency of adverse events were important factors among the healthcare workers.

Regarding COVID-19 vaccine acceptance and hesitancy, the distribution of responses (on the 5-point Likert scale) to the following two questions was similar between the groups: “When a vaccine for COVID-19 becomes available, I will get vaccinated” and “When everyone gets a vaccine for COVID-19, I will get vaccinated” (Table 2). The COVID-19 vaccine acceptance rates among the different groups were as follows: the general population, 48.6%; doctors, 54.2%; nurses, 45.5%; pharmacists, 49.6%; physiotherapists/occupational therapists, 49.6%; and medical clerks, 40.7%. The COVID-19 vaccine hesitancy rates were as follows: general population, 17.5%; doctors, 11.7%; nurses, 18.5%; pharmacists, 17.1%; physical/occupational therapists, 13.8%; and medical clerks, 21.5%.

Factors associated with “hesitant” and “not sure” responses regarding COVID-19 vaccination are shown in Table 3. After multivariable analysis, female sex, younger age, being a nurse, influenza vaccine hesitancy, and deciding on whether to be vaccinated based on concern for adverse events were positively associated with COVID-19 vaccine hesitancy. Current smoking, deciding on whether to be vaccinated based on vaccination fee, vaccine effectiveness, and epidemic situation in the area were negatively associated with COVID-19 vaccine hesitancy. Similarly, female sex, younger age, influenza vaccine hesitancy, and deciding on whether to be vaccinated based on concern for adverse events were positively associated with “not sure” responses to COVID-19 vaccination. Vaccine effectiveness was negatively associated with “not sure” responses to COVID-19 vaccination. However, being a nurse and being a current smoker were not associated with a “not sure” response. Instead, being a doctor was positively associated with a “not sure” response. Having children and deciding on whether to be vaccinated based on the duration of vaccine effectiveness were negatively associated with a “not sure” response.

COVID-19 vaccine acceptance was also compared between the general population and the healthcare workers under hypothetical vaccine effectiveness, duration of effectiveness, and adverse event frequencies (Figure 1). COVID-19 vaccine acceptance was high with higher vaccine effectiveness, a longer duration of effectiveness, and a lower frequency of adverse events. When the hypothetical vaccine effectiveness was 90%, there was approximately 80% acceptance in either group; however, when it was 50–70%, the acceptance rate differed by group. For instance, when the vaccine effectiveness rate was set at 50%, the acceptance rates among the general population and doctors were 22.8% and 36.7%, respectively. Acceptance rates for hypothetical durations of vaccine effectiveness were not different between the groups. Regardless of adverse event severity, the acceptance rate increased with decreasing adverse event frequency. The acceptance rate was low when the frequency was high, even for mild adverse events, such as influenza and high fever. The acceptance rate among doctors was high for any frequency of adverse events.

As shown in Table 4, vaccine confidence and literacy were higher among healthcare workers than among the general population. Age- and sex-adjusted scores were highest among doctors and lowest among the general population. Interestingly, compared with the general population, a greater proportion of healthcare workers were concerned about the adverse events of the newly developed vaccines and worried about serious adverse events.

## 4. Discussion

This was the first study to compare COVID-19 vaccine acceptance and hesitancy between the general population and healthcare workers in Japan. We found that nurses are 1.4 times more likely to be hesitant to the COVID-19 vaccine as compared to the general population.

Previous studies have reported COVID-19 vaccine acceptance rates of approximately 70% in the United States [4], 80% in Australia [22], 80% in England and Denmark, 70% in Italy and Germany, and 60% (lowest) in France [3]. A recent meta-analysis found an acceptance rate of approximately 70% [6]. The COVID-19 vaccine acceptance rate in our study was approximately 50%, which was lower than that reported in Western countries. On the other hand, the COVID-19 vaccine hesitancy rate was reported to be 14% in the United States [4], 5.8% in Australia [22], and 5–10% in European countries [3]. The COVID-19 vaccine hesitancy rate in our study was approximately 18%, which was higher than that in Western countries. This may reflect the fact that Japan is one of the countries with the strongest vaccination hesitancies in the world [10].

Several web-based cross-sectional surveys on COVID-19 vaccine acceptance and hesitancy have been reported in Japan [11,12,13,14,23]. In September 2020, Yoda et al. conducted a survey on COVID-19 vaccine preferences among 1100 people living in Tokyo and Osaka. They found acceptance and hesitancy rates of 65.7% and 12.3%, respectively [11]. According to a survey on COVID-19 vaccine preference conducted by Machida et al. in January 2021 among 2956 people living in Tokyo and metropolitan areas, the proportion of acceptance was 62.1% [13]. COVID-19 vaccine acceptance rates were higher in studies conducted in urban areas. Responses from residents in urban areas may have been influenced by the pandemic. In addition, the timing of our survey may have influenced the acceptance rate. A systematic review conducted before the initiation of vaccinations reported that the COVID-19 vaccine acceptance rate has declined over time since the start of the pandemic [6]. The COVID-19 vaccine hesitancy rate was higher in our study than in previous studies in Japan. A possible reason for this is the impact of media coverage, including social media coverage. In December 2020, vaccination was started in the United States, and anaphylaxis after vaccination was widely reported. In January 2021, the Japanese government announced that vaccination would start in late February in Japan. On January 17, two days before our survey, 29 older adults who received the COVID-19 vaccine were reported to have died in Norway. This might have increased public concern about the safety of COVID-19 vaccines. After our survey, the COVID-19 vaccine was approved for use in Japan on 14 February 2021. Subsequently, priority vaccination of healthcare professionals was initiated on 17 February 2021. Three nationwide web-based surveys were conducted after the introduction of the COVID-19 vaccine in Japan. Kadoya et al. [14] reported that the COVID-19 vaccine acceptance and hesitancy rates were 46.7% and 22.0%, and Nomura et al. [18] reported 56.1% and 11.0%, respectively. Okubo et al. reported that 88.7% of study participants chose “I want to be vaccinated” or “I want to be vaccinated after seeing how it goes”, and 11.3% chose “I don’t want to be vaccinated”. [12,14,23]. These findings were consistent with those of our study.

Female sex, younger age, being a nurse, influenza vaccine hesitancy, and concern about the frequency of adverse events were positively associated with vaccine hesitancy in our study. These factors have often been reported in previous studies [7,18]. Smokers were generally considered to be more likely to hesitate to vaccinate due to their lower awareness of health prevention behaviors. Unexpectedly, current smokers had a lower odds ratio of vaccine hesitancy in our study. Okubo et al. reported similar findings in their study [12]. This might be because of the fact that current smokers are at a higher risk of severe COVID-19, as is well publicized in the press. Similar to a previous study conducted in Israel [7], our study showed a higher odds ratio of vaccine hesitancy among nurses. This finding was independent of factors associated with vaccine hesitancy, such as sex and age. In addition, multivariable analysis by sex revealed that nurses were tended to be associated with hesitancy, although statistical significance was not detected due to limited sample size (Appendix A). Confidence in immunization, in general, tended to be higher among healthcare workers than among the general population; however, there were more healthcare workers worried about vaccine safety. Takamatsu et al. examined healthcare institutions in the Japanese metropolitan area prior to the introduction of the COVID-19 vaccine and reported a lower COVID-19 vaccine acceptance rate among nurses [24]. However, the study included only healthcare workers. Clearly, healthcare professionals, especially nurses, need to improve their immunization knowledge since they are important in educating the general population on the safety and effectiveness of immunization.

The vaccine acceptance under hypothetical vaccine effectiveness and adverse events was previously examined in the U.S. [19] and Japan [25]. Kreps et al. conducted their survey in June 2020, prior to the initiation of vaccination in the United States [19], and Kawata et al. conducted their survey from February 16 to March 15, 2021, after the initiation of vaccination in Japan [25]. The COVID-19 vaccine acceptance rate was approximately 30% in our study and 42% in the Japanese survey by Kawata et al. [25], when the occurrence of serious adverse events, such as hospitalization and death, was one person per 10,000 vaccinations. These rates were lower than the 55% reported in the U.S. survey [19]. The COVID-19 vaccine acceptance rate (60%) in the case of serious adverse events in one person per one million vaccinations in our study was similar to that in the U.S. survey. This may indicate that Japanese people are less tolerant of adverse events. On the other hand, when hypothetical mild adverse events, such as influenza or high fever, occurred in one dose per 10 doses, the vaccine acceptance rate was 27% in our study, 48% in Kawata’s survey [25], and 55% in the U.S. survey [19]. It is considered that information on the degree and frequency of adverse reactions had not fully reached the general population compared to healthcare workers at the time of our survey. However, it had been reported that approximately half of the people vaccinated with the Pfizer vaccine developed fever and malaise. The COVID-19 vaccine acceptance rates when hypothetical vaccine effectiveness was set at 90%, 70%, and 50% were, respectively, 80%, 54%, and 23% in our study; this compares to 58%, 51%, and 41% in the study by Kawata et al., and 61%, 56%, and 51% in the U.S. study. These results suggest that the Japanese population tends to seek higher effectiveness and safety of the COVID-19 vaccine than the U.S. population. Accordingly, public health policymakers must make efforts to increase confidence and literacy in the vaccine. Galle et al. reported that 91.9 of undergraduate students in Italy were keen to receive a COVID-19 vaccination since the Italian Ministry of Health had launched a national vaccination campaign to counteract the COVID-19 pandemic [26]. Better communication regarding the risks and benefits of vaccination is needed.

The strength of the present study is that we compared vaccine acceptance and hesitancy between the general population and healthcare workers who are influential regarding vaccine acceptance in the general population. We must acknowledge that this study had several limitations. First, web-based surveys tend to have selection and sampling biases. The survey participants may comprise regular Internet users; thus, they might have a higher socioeconomic status, which influences vaccine perception. However, most surveys on COVID-19 vaccine perception have used the same method; thus, our results are comparable to those of other studies. In addition, the degree of interest in the COVID-19 vaccine might not have influenced participation because participants received points, which could be used to purchase products and services from partner companies after completing the survey. Second, this was a cross-sectional study; therefore, causal relationships could not be established. However, our purpose was not to establish causality but to assess the factors associated with vaccine acceptance and hesitancy. Lastly, this study was conducted before the initiation of COVID-19 vaccinations in the population, and vaccine perception may change over time. In addition, the survey was conducted prior to the Delta variant outbreak in Japan, which may propel people to become vaccinated because it is highly contagious; therefore, the time at which the survey was conducted is an important consideration. Despite these limitations, to the best of our knowledge, this is the first study to compare COVID-19 vaccine acceptance and hesitancy and their related factors between the general population and healthcare workers in Japan.

## 5. Conclusions

The COVID-19 vaccine acceptance rate was 48.6%, and the hesitancy rate was 17.5% among the Japanese general population in January 2021. Nurses showed lower COVID-19 vaccine acceptance and higher hesitancy. Similar to previous studies, a higher proportion of women and young people, those who did not receive the influenza vaccine, and those who cited the frequency of adverse reactions as a factor influencing their decision on vaccination showed vaccination hesitancy. Even after adjusting for the influence of these factors, nurses are 1.4 times more likely to be hesitant as compared to the general population. Confidence in immunization in general was higher among healthcare workers than among the general population. However, safety concerns were greater among healthcare workers. Interventions to improve immunization literacy are needed among both the general population and healthcare workers.

## Figures and Tables

**Figure 1 vaccines-09-01389-f001:**
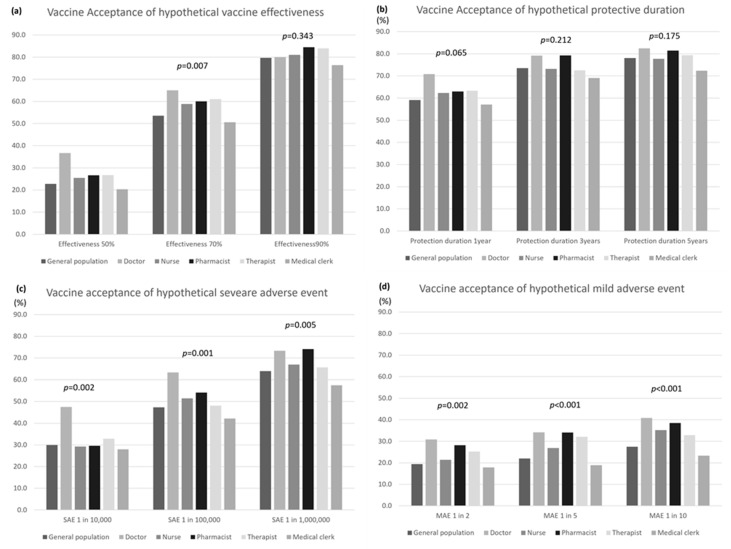
COVID-19 vaccine acceptance under hypothetical vaccine effectiveness, duration of effectiveness, and frequency of adverse events. (**a**) Vaccine acceptance under hypothetical vaccine effectiveness; (**b**) vaccine acceptance under hypothetical durations of effectiveness; (**c**) vaccine acceptance under hypothetical frequency of severe adverse events (SAE); (**d**) vaccine acceptance under hypothetical frequency of mild adverse events (MAE).

**Table 1 vaccines-09-01389-t001:** Characteristics of study participants.

Characteristic		General Population (*n* = 6180)		Healthcare Workers (*n* = 1030)	
		*n*	%	*n*	%
Sex	Women	3090	50.0	725	70.4
Age group (years)	20–29	1030	16.7	80	7.8
	30–39	1030	16.7	282	27.4
	40–49	1030	16.7	326	31.7
	50–59	1030	16.7	244	23.7
	60–69	2060	33.3	98	9.5
State of emergency *		4036	65.3	561	54.5
Healthcare workers	Doctor			120	11.7
	Nurse			369	35.8
	Pharmacist			135	13.1
	Physical /occupational therapist			131	12.7
	Medical clerk			275	26.7
Marital status	Married	3836	62.1	665	64.6
Have children		3561	57.6	625	60.7
Annual household income	<4 million yen	1736	28.1	140	13.6
	≥4 million yen	2996	48.5	636	61.7
	Not sure	1448	23.4	254	24.7
Educational level	High school graduate	2010	32.5	108	10.5
	College or more	4170	67.5	922	89.5
Obesity (BMI ≥ 30)		1239	20.0	172	16.7
Comorbidity	Present	2049	33.2	302	29.3
Smoking status	Current smoker	1038	16.8	113	11.0
Having a history of influenza vaccination					
2020/21 season	Present	2867	46.4	872	84.7
2019/20 season	Present	2566	41.5	857	83.2
Age		2935	47.5	503	48.8
Comorbidity		3671	59.4	645	62.6
Occupation		1732	28.0	560	54.4
Vaccination fee		1906	30.8	239	23.2
Doctors’ recommendation		1257	20.3	160	15.5
Vaccine effectiveness		3719	60.2	673	65.3
Duration of vaccine effectiveness		2372	38.4	481	46.7
Frequency of adverse events		3698	59.8	680	66.0
Family recommendation		275	4.4	48	4.7
COVID-19 epidemic situation		2057	33.3	361	35.0

* A state of emergency was declared in Tokyo, Chiba, Saitama, Kanagawa, Tochigi, Gifu, Aichi, Kyoto, Osaka, Hyogo, and Fukuoka.

**Table 2 vaccines-09-01389-t002:** Perception of COVID-19 vaccination.

		General Population (*n* =6180)	Doctor (*n* = 120)	Nurse (*n* = 369)	Pharmacist (*n* = 135)	Therapist (*n* = 131)	Medical Clerk (*n* = 275)	*p*-Value *
		*n*(%)	*n*(%)	*n*(%)	*n*(%)	*n*(%)	*n*(%)	
	When a vaccine for COVID-19 becomes available, I will get vaccinated							
Hesitant	Strongly disagree	339(5.5)	6(5.0)	25(6.8)	7(5.2)	4(3.1)	26(9.5)	0.069
	Disagree	743(12.0)	8(6.7)	43(11.7)	16(11.9)	14(10.7)	33(12.0)	
Not sure	Neither agree nor disagree	2095(33.9)	41(34.2)	133(36.0)	45(33.3)	48(36.6)	104(37.8)	
Acceptance	Agree	2164(35.0)	40(33.3)	114(30.9)	46(34.1)	52(39.7)	89(32.4)	
	Strongly agree	839(13.6)	25(20.8)	54(14.6)	21(15.6)	13(9.9)	23(8.4)	
	I will get vaccinated when everyone gets a vaccine for COVID-19							
Hesitant	Strongly disagree	259(4.2)	4(3.3)	21(5.7)	6(4.4)	3(2.3)	18(6.5)	0.125
	Disagree	559(9.0)	7(5.8)	34(9.2)	12(8.9)	13(9.9)	20(7.3)	
Not sure	Neither agree nor disagree	1665(26.9)	37(30.8)	109(29.5)	37(27.4)	34(26.0)	90(32.7)	
Acceptance	Agree	2719(44.0)	50(41.7)	145(39.3)	59(43.7)	70(53.4)	117(42.5)	
	Strongly agree	978(15.8)	22(18.3)	60(16.3)	21(15.6)	11(8.4)	30(10.9)	

* The chi-squared test was used to analyze the differences between the general population and healthcare workers.

**Table 3 vaccines-09-01389-t003:** Factors associated with vaccine hesitancy.

		Hesitant (*n* = 1264)				Not sure (*n* = 2466)			
		Crude		Adjusted ***		Crude		Adjusted ***	
		OR	(95%CI)	OR	(95%CI)	OR	(95%CI)	OR	(95%CI)
Sex	Women	1.55	(1.36–1.76)	1.57	(1.35–1.82)	1.17	(1.02–1.34)	1.53	(1.63–1.72)
Age group (years)	20–29	1.83	(1.49–2.25)	1.39	(1.08–1.80)	1.31	(1.11–1.54)	1.01	(0.83–1.24)
	30–39	2.11	(1.74–2.56)	2.01	(1.60–2.53)	1.68	(1.44–1.96)	1.49	(1.25–1.78)
	40–49	2.03	(1.67–2.47)	1.80	(1.44–2.24)	1.67	(1.44–1.95)	1.43	(1.20–1.69)
	50–59	1.60	(1.31–1.95)	1.45	(1.16–1.0)	1.36	(1.16–1.58)	1.19	(1.01–1.41)
	60–69	1.00	(reference)	1.00	(reference)	1.00	(reference)	1.00	(reference)
State of emergency *		1.12	(0.98–1.28)	1.09	(0.94–1.25)	1.04	(0.93–1.16)	1.03	(0.92–1.15)
Population	General	1.00	(reference)	1.00	(reference)	1.00	(reference)	1.00	(reference)
	Doctor	0.60	(0.34–1.07)	1.23	(0.67–2.26)	0.90	(0.61–1.34)	1.56	(1.03–2.37)
	Nurse	1.12	(0.84–1.50)	1.39	(1.00–1.92)	1.14	(0.90–1.43)	1.21	(0.94–1.56)
	Pharmacist	0.95	(0.59–1.54)	1.27	(0.76–2.11)	0.96	(0.66–1.41)	1.15	(0.77–1.71)
	Therapist	0.77	(0.45–1.30)	1.31	(0.88–1.94)	1.06	(0.73–1.54)	1.31	(0.88–1.94)
	Medical clerk	1.46	(1.06–2.02)	1.39	(0.98–1.98)	1.33	(1.01–1.75)	1.18	(0.89–1.58)
Marital status	married	0.72	(0.63–0.82)	0.98	(0.82–1.17)	0.81	(0.72–0.90)	0.92	(0.80–1.06)
Have children		0.65	(0.57–0.74)	0.92	(0.82–1.17)	0.79	(0.71–0.88)	0.98	(0.85–1.14)
House income	≥4 million yen	0.83	(0.73–0.95)	0.88	(0.76–1.02)	0.87	(0.78–0.97)	0.93	(0.83–1.04)
Educational level	College or more	1.06	(0.92–1.23)	1.10	(0.94–128)	0.94	(0.84–1.06)	0.97	(0.86–1.09)
Obesity (** BMI ≥ 30)		0.79	(0.66–0.94)	0.89	(0.74–1.07)	0.88	(0.78–1.00)	1.03	(0.89–1.18)
Comorbidity	Present	0.68	(0.59–0.78)	0.99	(0.85–1.16)	0.65	(0.58–0.73)	0.80	(0.71–1.16)
Current smoker		0.72	(0.60–0.87)	0.69	(0.57–0.85)	0.95	(0.82–1.09)	0.94	(.82–1.10)
Having history of influenza vaccination									
2020/21 season	Present	2.12	(2.21–2.87)	2.27	(1.87–2.75)	1.82	(1.64–2.02)	1.65	(1.42–1.92)
2019/20 season	Present	2.17	(1.90–2.48)	1.38	(1.14–1.68)	1.66	(1.50–1.84)	1.25	(1.08–1.46)
Factors to decide whether to take COVID-19 vaccine									
Age		0.56	(0.49–0.64)	0.94	(0.78–1.12)	0.67	(0.61–0.75)	0.82	(0.71–0.94)
Comorbidity		0.53	(0.47–0.61)	0.71	(0.59–0.84)	0.73	(0.66–0.82)	0.92	(0.80–1.06)
Occupation		0.65	(0.56–0.75)	0.87	(0.73–1.04)	0.94	(0.75–0.94)	0.97	(0.84–1.10)
Vaccination fee		0.58	(0.50–0.68)	0.60	(0.51–0.72)	0.92	(0.82–1.03)	0.89	(0.79–1.01)
Doctors’ recommendation		0.57	(0.47–0.68)	0.83	(0.68–1.00)	0.79	(0.69–0.89)	0.93	(0.81–1.07)
Vaccine effectiveness		0.60	(0.53–0.68)	0.62	(0.53–0.74)	0.90	(0.81–1.00)	0.85	(0.74–0.97)
Duration of vaccine effectiveness		0.58	(0.50–0.66)	0.75	(0.63–0.89)	0.79	(0.71–0.87)	0.78	(0.68–0.89)
Frequency of adverse event		1.08	(0.94–1.23)	2.02	(1.71–2.38)	1.24	(1.11–1.38)	1.64	(1.44–1.87)
Families’ recommendation		0.44	(0.30–0.64)	0.76	(0.51–1.13)	0.63	(0.49–0.81)	0.77	(0.59–1.01)
Epidemic situation of COVID-19		0.58	(0.50–0.67)	0.76	(0.64–0.89)	0.87	(0.78–0.97)	0.97	(0.85–1.09)

* A state of emergency was declared in Tokyo, Chiba, Saitama, Kanagawa, Tochigi, Gifu, Aichi, Kyoto, Osaka, Hyogo, and Fukuoka. ** BMI: Body mass index. *** Adjusted for variables listed in the table.

**Table 4 vaccines-09-01389-t004:** Assessment of vaccine confidence and literacy (proportion of participants who chose “strongly agree” or “agree”.).

	General Population	Doctor	Nurse	Pharmacist	Therapist	Medical Clerk	*p*-Value
	(*n* = 6180)	(*n* = 120)	(*n* = 369)	(*n* = 135)	(*n* = 131)	(*n* = 275)	
1 Vaccines are important for my health	59.1	68.3	67.8	74.1	62.6	59.3	<0.001
2 Vaccines are effective	63.8	72.5	68.8	72.6	67.9	62.5	0.028
3 My vaccination is important for the health of others in my community	61.0	69.2	64.0	71.1	55.7	59.3	0.032
4 New vaccines carry more risks than older vaccines (R)	44.1	36.7	53.1	59.3	42.7	50.9	<0.001
5 I am concerned about serious adverse effects of vaccines (R)	67.7	46.7	69.1	75.6	67.9	73.8	<0.001
6 I do not need vaccines for diseases that are not common anymore (R)	13.3	8.3	12.7	10.4	13.0	9.8	0.291
7 Vaccines are safe	22.0	29.2	21.1	25.2	23.7	16.4	0.079
8 Serious adverse reactions may occur due to the vaccination (R)	57.3	78.3	66.9	77.0	56.5	63.3	<0.001
9 I have difficulty getting immunized (no time, far medical institutions, etc.) (R)	22.1	15.8	17.6	25.2	12.2	14.5	<0.001
10 We do not necessary to take voluntary vaccination (R)	22.2	11.7	17.6	14.8	16.8	17.5	0.001
11 I do not take vaccine, if everyone around me is immunized (R)	8.7	10.0	9.2	11.9	9.2	5.8	0.42
12 It is easy to obtain correct information on immunization	23.5	39.2	35.2	42.2	23.7	27.3	<0.001
13 It is easy to understand why immunization is needed.	49.4	67.5	55.8	61.5	51.9	51.6	<0.001
14 I have been able to accurately understand the vaccinations I have received	34.5	70.0	51.8	57.8	42.7	37.5	<0.001
Sex and age adjusted score. Mean (95% confidence intervals)	43.9 (43.7–44.0)	47.3 (46.2–48.3)	46 (45.3–46.6)	45.9 (44.9–47.0)	45.5 (44.4–16.5)	45.0 (44.3–45.7)	<0.001

To build the score, answers were allocated the following scores: strongly disagree = 1, disagree = 2, neither agree nor disagree = 3, agree= 4, strongly agree = 5. R means reverse allocation.

## Data Availability

The data presented in this study are available on request from the corresponding author (M.H.). The data are not publicly available due to privacy concerns.

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
