# Peer review of "Differences in COVID-19 Vaccine Acceptance, Hesitancy, and Confidence between Healthcare Workers and the General Population in Japan"

_vaccines, 2021, doi:10.3390/vaccines9121389_

Round 1

Reviewer 1 Report

Major comments

1. The aim of this study was not clear. For example, in the introduction the authors mentioned “This study aimed to investigate COVID-19 vaccine acceptance and hesitancy, immunization confidence in general, and associated factors among healthcare workers and the general Japanese population”.

However, in the first paragraph of the discussion, the authors mentioned “This was the first study to compare COVID-19 vaccine acceptance and hesitancy between the general population and healthcare workers in Japan.” What would be the aim of this study?

The author also needs to articulate the aim in the abstract.

2. There have been many variables and data but the analysis was not proper.

For the general population, the authors may be able to show the analysis by gender in logistics regression since gender is one determinant. The results can be shown in the appendix or text.

3. More detailed information would be necessary on factors related to 2.2.4. As the authors show some data at the bottom of table 1. However, we are not able to understand what it means.

4. The authors articulated the hesitancy among nurses but nurses were not significantly associated with hesitancy in crude analysis. It may have interaction possibly with “female”. Thus, I am not sure if the authors can emphasize the hesitancy among nurses based on this result.

Author Response

Point 1: The aim of this study was not clear. For example, in the introduction the authors mentioned “This study aimed to investigate COVID-19 vaccine acceptance and hesitancy, immunization confidence in general, and associated factors among healthcare workers and the general Japanese population”.   

However, in the first paragraph of the discussion, the authors mentioned “This was the first study to compare COVID-19 vaccine acceptance and hesitancy between the general population and healthcare workers in Japan.” What would be the aim of this study?

The author also needs to articulate the aim in the abstract.

Response 1: We appreciate this critical comment. We corrected the aim in the introduction (lines 69-71) and added it in the abstract (lines 14,15).

Point 2: There have been many variables and data but the analysis was not proper.

For the general population, the authors may be able to show the analysis by gender in logistics regression since gender is one determinant. The results can be shown in the appendix or text.

Response 2: We appreciate this valuable comment. According to this reviewer, we examined factors associated with vaccine hesitancy by gender, and show the results in Appendix and text (line 288-290).

Point 3:  More detailed information would be necessary on factors related to 2.2.4. As the authors show some data at the bottom of table 1. However, we are not able to understand what it means.

Response 3: We corrected this section to make it clear what we did. Participants chose all of the factors that applied to them as important factors in deciding whether or not to vaccinate in listed factors in follows: age, comorbidity, occupation vaccination fee, doctors’ recommendation, vaccine effectiveness, duration of effectiveness, frequency of adverse events, family recommendation, and the COVID-19 epidemic situation (lines 139-140, 146, 149-152).

Point 4:. The authors articulated the hesitancy among nurses but nurses were not significantly associated with hesitancy in crude analysis. It may have interaction possibly with “female”. Thus, I am not sure if the authors can emphasize the hesitancy among nurses based on this result.

Response 4: Although nurses were not significantly associated with hesitancy in crude analysis, multivariable analysis that adjusted for confounding factors including sex shows statistically significant association between them (Table 3). In addition, multivariable analysis by gender revealed that nurses were tended to be associated with hesitancy, although statistical significance were not detected due to limited sample size (Appendix A). Therefore, we believe that nurses are more likely to be hesitant as compared to the general population.

Reviewer 2 Report

The study looks at the rates of COVID vaccine hesitancy among health workers and the general population, and tries to compare the rates and likelihood of hesitancy among these two populations.

The topic is timely and highly relevant considering that many countries are grappling with the issue of hesitancy against COVID vaccine. There are lots of relevant lessons for the country of study ( Japan, as well as for developed and developing countries in terms of how to monitor for hesitancy, and how to programmatically approach vaccine hesitancy. Health workers are critical players in COVID vaccine roll out, and the presence of strong hesitancy in this group, sis likely to affect the vaccine acceptance among the general population.

The study provides an recent measure ( in quantitative terms) of COVID vaccine hesitancy in different population groups in Japan, and compares it to previous studies of vaccine hesitancy in the same country as well as in a host of other countries. The critical point of hesitancy among health workers is key here.

I thought that the methodology was good. The limitations of the study in terms of selection of the study participants has been clearly stated. Otherwise, doing  similar studies among the general population, using a different approach to selecting samples would improve the study outcome and validity of the results.

I think this is a highly relevant research study on COVID vaccine acceptance and hesitancy, and provides very good information to help improve the uptake of the vaccines in Japan. The background , methodology , results and conclusions are very well described. The tables are well labelled and complement the information in the results section. The discussion section compares these findings with other recent studies from Japan and other countries. One minor comment I have is on lines 233 - 234 and lines 341 - 342 where the statement needs to be rephrased to show that "nurses are 1.4 times more likely to be hesitant as compared to the general population" as compared to the current text which mistakenly states that "nurses showed 1.4 times more hesitancy than that of the general population". I note from the data that the proportion of hesitancy is not much different, but the likelihood ( Odds ratio) was more among nurses and this was statistically significant.

Author Response

Point 1: One minor comment I have is on lines 233 - 234 and lines 341 - 342 where the statement needs to be rephrased to show that "nurses are 1.4 times more likely to be hesitant as compared to the general population" as compared to the current text which mistakenly states that "nurses showed 1.4 times more hesitancy than that of the general population". I note from the data that the proportion of hesitancy is not much different, but the likelihood ( Odds ratio) was more among nurses and this was statistically significant.

Response 1: We appreciate this comment. We corrected these sentences according to this reviewer’s comment on lines 238-240, and 353-354.

Reviewer 3 Report

First of all, I would like to thank for the opportunity to review this paper. COVID-19 is an ongoing pandemic that has resulted in global health, economic and social crises. Actually, the vaccination campaign is the first method to counteract the COVID-19 pandemic; however, sufficient vaccination coverage is conditioned by the people’s acceptance of these vaccines in the general population and health care workers. In this context, the paper under review is aimed at investigating COVID-19 vaccine acceptance and hesitancy and associated factors among healthcare workers and the general population in Japan.

The article is interesting and may provide interesting information for public health, but it must be improved especially underling the results useful internationally and not only in Japan.

Introduction: The authors should make more clear what is the gap in the literature that is filled with this study? What is the international situation regarding the acceptance of the vaccination in the adult population (refer to the article with DOI: https://doi.org/10.3390/vaccines9060638) What is the possible international contribution of the study to the literature? What are the implications of the study?

Methods: The enrolment procedure must be better specified, who was involved in the survey? How did the authors choose the way used to send the questionnaire? Who is registered to this panel of a web survey? How did you avoid the selection bias? What is the reference population? About the questionnaire, no mention to a validation process is reported. What about face validity, reliability and intelligibility?

Statistical analysis: I suggest to insert a measure of the magnitude of the effect for the comparisons. Please consider to include effect sizes.

Discussion: I also suggest expanding. Emphasize the contribution of the study to the literature. The authors reported that interventions to improve immunization literacy are needed; therefore, discuss the implications and recommendations based on previous experience in other population groups also reporting the effectiveness of the information strategy (refer to Gallè, F. et al Knowledge and Acceptance of COVID-19 Vaccination among Undergraduate Students from Central and Southern Italy. Vaccines 2021, 9, 638). Limits section can be improved.

Author Response

Point 1: The article is interesting and may provide interesting information for public health, but it must be improved especially underling the results useful internationally and not only in Japan.

Introduction: The authors should make more clear what is the gap in the literature that is filled with this study? What is the international situation regarding the acceptance of the vaccination in the adult population (refer to the article with DOI: https://doi.org/10.3390/vaccines9060638) What is the possible international contribution of the study to the literature? What are the implications of the study?

Response 1: We appreciate this construct advise. We mentioned the international situation regarding the acceptance of the vaccination in the adult population (lines 38-51), as well as situation in Japan (lines 58-69). However, the possible international contribution and implications of this study was not clear, as this reviewer pointed out. We added a sentence in below to lines 69-71.

Investigating vaccine acceptance and hesitancy between healthcare workers and general population in Japan might have lots of relevant lessons for developed and developing countries in terms of how to monitor or programmatically approach vaccine hesitancy

Point 2: Methods: The enrolment procedure must be better specified, who was involved in the survey? How did the authors choose the way used to send the questionnaire? Who is registered to this panel of a web survey? How did you avoid the selection bias? What is the reference population? About the questionnaire, no mention to a validation process is reported. What about face validity, reliability and intelligibility?

Response 2: We appreciate these critical comments. The survey participants were from voluntary registered as a panel of a web survey company (Macromill, Inc., Tokyo). The survey was explained by e-mail and study participants were enrolled using e-mails and apps until the target number was reached. To avoid selection bias for the degree of interest in the COVID-19 vaccine, participants received points, which could be used to purchase products and services from partner companies, after completing the survey (lines 76-92, 329-336).

Although we did not validate questionnaire, vaccine confidence and literacy were measured using 14 questions based on a validated scale or index. We selected six of the ten items of the Vaccine Hesitancy Scale, which was developed by the WHO Strategic Advisory Group of Experts and validated by Larson et al.. The remaining eight items related to vaccine confidence and literacy (lines 116-137).

Point 3: Statistical analysis: I suggest to insert a measure of the magnitude of the effect for the comparisons. Please consider to include effect sizes.

Response 3: Thank you for this suggestion. However, I do not agree this, because I think that it is not appropriate to mention for effect, because this study is cross-sectional design.

Point 4: Discussion: I also suggest expanding. Emphasize the contribution of the study to the literature. The authors reported that interventions to improve immunization literacy are needed; therefore, discuss the implications and recommendations based on previous experience in other population groups also reporting the effectiveness of the information strategy (refer to Gallè, F. et al Knowledge and Acceptance of COVID-19 Vaccination among Undergraduate Students from Central and Southern Italy. Vaccines 2021, 9, 638). Limits section can be improved.

Response 4: I appreciate this valuable suggestion. I added a sentence in below to the discussion section (lines 322-324). 

Galle et al reported that 91.9 of undergraduate student in Italy were keen to receive a COVID-19 vaccination, since Italian Ministry of Health had launched a national vaccination campaign to counteract the COVID-19 pandemic.

Round 2

Reviewer 3 Report

The authors improved the manuscript according to the comments provided, the article - in my opinion - is suitable for publication